# Drug discovery of small molecules targeting the higher-order hTERT promoter G-quadruplex

Robert C. Monsen[1], Jon M. Maguire[1], Lynn W. DeLeeuw[1], Jonathan B. Chaires[1,2,3]*, John O. Trent[1,2,3]*

1 UofL Health Brown Cancer Center, University of Louisville, Louisville, Kentucky, United States of America,
2 Department of Medicine, University of Louisville, Louisville, Kentucky, United States of America,
3 Department of Biochemistry and Molecular Genetics, University of Louisville, Louisville, Kentucky, United States of America

* j.chaires@louisville.edu (JBC); john.trent@louisville.edu (JOT)

**Data Availability Statement:** All relevant data are within the article and its Supporting information files.

## Abstract

DNA G-quadruplexes (G4s) are now widely accepted as viable targets in the pursuit of anti-cancer therapeutics. To date, few small molecules have been identified that exhibit selectivity for G4s over alternative forms of DNA, such as the ubiquitous duplex. We posit that the lack of current ligand specificity arises for multiple reasons: G4 atomic models are often small, monomeric, single quadruplex structures with few or no druggable pockets; targeting G-tetrad faces frequently results in the enrichment of extended electron-deficient polyaromatic end-pasting scaffolds; and virtual drug discovery efforts often under-sample chemical search space. We show that by addressing these issues we can enrich for non-standard molecular templates that exhibit high selectivity towards G4s over other forms of DNA. We performed an extensive virtual screen against the higher-order hTERT core promoter G4 that we have previously characterized, targeting 12 of its unique loop and groove pockets using libraries containing 40 million drug-like compounds for each screen. Using our drug discovery funnel approach, which utilizes high-throughput fluorescence thermal shift assay (FTSA) screens, microscale thermophoresis (MST), and orthogonal biophysical methods, we have identified multiple unique G4 binding scaffolds. We subsequently used two rounds of catalogue-based SAR to increase the affinity of a disubstituted 2-aminoethyl-quinazoline that stabilizes the higher-order hTERT G-quadruplex by binding across its G4 junctional sites. We show selectivity of its binding affinity towards hTERT is virtually unaffected in the presence of near-physiological levels of duplex DNA, and that this molecule downregulates *hTERT* transcription in breast cancer cells.

## Introduction

Nucleic acids have the capacity to form multiple types of tertiary structure, such as, but not limited to, duplex (B-, A-, and Z-DNA), triplex, and quadruplex. Genomic regions with high guanine content, specifically with multiple runs of consecutive guanines, can form highly

**Funding:** This study was supported in the form of funding by the National Institutes of Health (NIH) [GM077422] awarded to JBC and JOT. Funding for publication was from the UofLHealth Brown Cancer Center. The funders had no role in study design, data collection and analysis, decision to publish, or preparation of the manuscript.

**Competing interests:** The authors have declared that no competing interests exist.

stable structures known as G-quadruplexes (G4s) [1]. G4s are composed of stacked planar layers of guanine nucleotides cyclically Hoogsteen H-bonded to form G-tetrads. Analyses of the human genome have revealed between $3x10^5$ to $7x10^5$ potential G4 forming regions [2, 3], and as much as 40% of these putative quadruplex-forming sequences (PQS) reside in gene promoters [4]. With the advent of fluorescent G4-specific small molecules and G4-specific antibodies, there is now *in situ* evidence of G4 formation in cellular DNA [5–7]. Importantly, promoter G4s are abundant and evolutionarily conserved in oncogenes [3, 8] and can impose proximal regulatory effects on transcription of adjacent genes [9, 10]. A compelling example of G4-mediated gene modulation is the repression of *c-MYC* using G4-targeting small molecules [9, 11]. While this has energized research targeting the promoter G4s of "undruggable" proteins [12], the issue of ligand specificity remains [10, 13, 14].

To date, most G4 drug discovery has been targeting the small (<12 kDa), monomeric, single quadruplex structures, that are often modified from the native biological sequences to make them most amenable to NMR and X-ray diffraction studies. Many promoter G4s have been characterized at the atomic level, such as *c-MYC* [15], *KRAS* [16], *HRAS* [17], *HIF* [18], and *VEGF* [19], permitting the use of structure-based drug discovery [14]. These efforts frequently result in the enrichment of extended electron-deficient polyaromatic molecules that preferentially end-paste to the terminal tetrad faces rather than binding with specificity to loops or grooves [20]. The maximization of π-π or cation-π interactions leads to a strong interaction, but low selectivity [13]. In some instances, this non-specificity can be improved by adding or altering constituent groups or sidechains to increase favorable interactions with distinctive loop or groove regions [20–23]; however, this comes at a price, as the modified molecules tend to deviate from "drug-likeness" the more they are altered (i.e. MW increases, and deviation from Lipinski's rule of 5) [24, 25].

An alternative approach for increasing specificity towards G4s is to target higher-order G4 structures [14, 20]. G-quadruplexes can form higher-order structure through sandwiching of flanking nucleotides between consecutive G4s or by direct stacking of terminal G-tetrads [26, 27]. The allure of targeting higher-order G4s is that they offer a richer drug targeting landscape among their unique G4-junctions, loops, and grooves [14, 28, 29] that is quantifiably more "druggable" [30]. Unfortunately, higher-order G4s have proven difficult to characterize by traditional methods (NMR and/or X-ray crystallography), so few have been structurally characterized [31, 32]. Recently, using an integrative structural biology (ISB) strategy, we have shown that the human telomerase reverse transcriptase (hTERT) core promoter folds into three contiguous parallel G4s with multiple unique drug targeting pockets [31, 33] (Fig 1).

The hTERT core promoter G4 is located at -168 to -100 relative to the transcription start site and has been extensively investigated by us [31, 33] and others [34–39]. hTERT, and its cognate RNA, form the ribonucleoprotein complex which is responsible for maintenance of the telomeres [40]. In normal somatic cells, hTERT activity is tightly regulated or entirely absent [41]. Under "normal" circumstances, dividing hTERT-negative cells will eventually experience telomere shortening which elicits a DNA damage response that ultimately results in senescence or cell death [42]. Forced re-expression of hTERT in hTERT-negative cell lines is sufficient to extend cellular replication [43, 44]. Knockdown of hTERT in cancer cell lines and tumor models results in reduced telomere maintenance, sensitization to chemotherapeutics, and in some cases, direct induction of apoptosis or senescence [45–47]. hTERT's criticality in maintaining most malignant cell types makes it an ideal cancer target. Contemporary techniques targeting telomerase, such as small molecule inhibitors, gene therapy, anti-sense oligonucleotides, and immunotherapies, have all shown that hTERT inhibition is a viable mechanism to inhibit cancer [48]. However, to date, no hTERT inhibitors have been successful clinically [49].

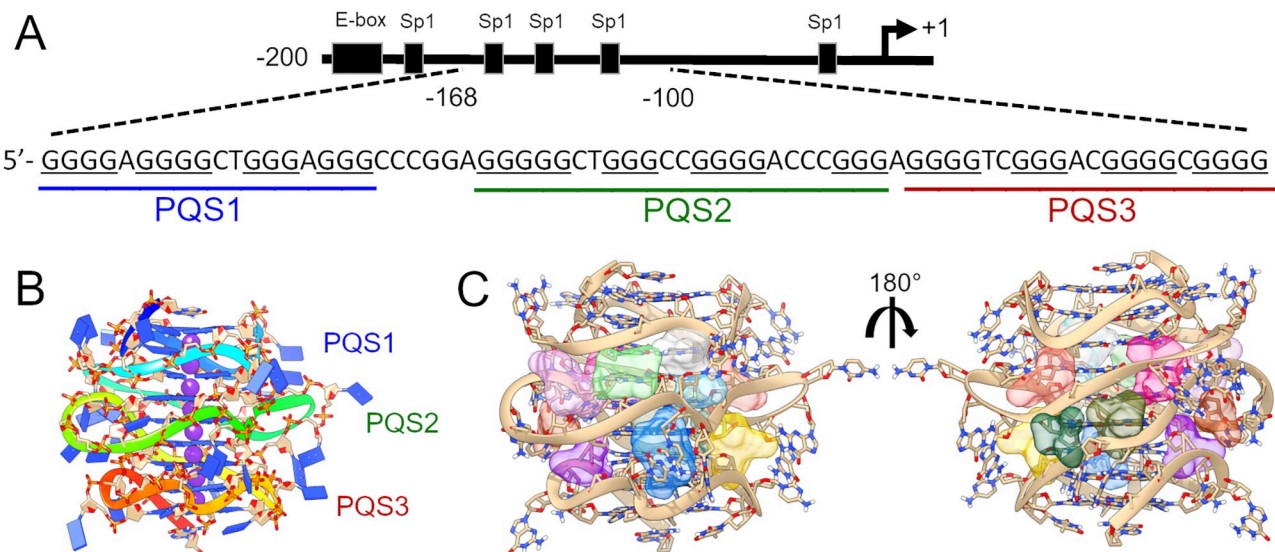

**Fig 1. The hTERT core promoter and higher-order G-quadruplex.** (A) The hTERT core promoter spanning from -200 to +1 with known Sp1 binding sites and Myc E-box motif represented schematically. The hTERT G4 region is displayed with putative quadruplex forming sites ('PQS') highlighted. (B) The hTERT higher-order G4 from refs [31, 33] with color coding from 5' (blue) to 3' (red), corresponding to the blue, green, and red underlined regions in A. (C) Surflex-Dock virtual screening sites represented as 12 space-filling protomols of various colors situated among the loops and grooves of the model in B.

The viability of inhibiting hTERT by targeting its higher-order G4 structure has been demonstrated previously using traditional *in vitro* screens with known G4 binding ligands. In the first report, Palumbo *et al.* demonstrated that both TMPyP4 and Telomestatin, two promiscuous G4 binding small molecules, could stabilize the hTERT quadruplex *in vitro* [34]. Independently, Micheli *et al.* reported that modified perylene diimide compounds could stabilize the higher-order hTERT G4 using *in vitro* Taq polymerase stop assays [36]. Later, Kang and colleagues identified an hTERT G4-stabilizing small molecule derivative of an acridine scaffold (known by its NCI designation of "GTC365") that stabilized the higher-order hTERT G4 *in vitro* and significantly reduced hTERT mRNA and protein levels in breast cancer cells [38]. We note that no atomistic model was proposed to explain the G4-ligand interaction in any of these cases, nor was evidence of selectivity for the hTERT G4 over other DNA topologies reported. Altogether, these studies provide evidence that stabilizing the higher-order hTERT quadruplex with small molecules is a viable route to indirect inhibition of telomerase.

Here we report the first virtual drug screening of the higher-order hTERT core promoter G4. We first apply our drug discovery funnel approach [50] to enrich for drug-like scaffolds using virtual screening from more than 40 million drug-like small molecule-containing libraries targeted at 12 sites identified among our higher-order hTERT model's G4 stacking interfaces (Fig 1C) [31]. Using fluorescent thermal shift assays (FTSA), we have identified multiple small molecules that bind with high selectivity over a duplex hairpin control sequence. We subsequently used a structure-activity relationship approach "SAR by catalogue" to identify seven more commercially available small analog molecules with greatly increased thermal stabilizing ability and specificity for G4 DNA over other forms. Direct binding interactions were confirmed by orthogonal methods (CD, MST, competition dialysis). One of these molecules, a disubstituted quinazoline, showed favorable selectivity in competition dialysis experiments for hTERT over monomeric single G4s, triplex, and duplex DNA. After a third round of SAR by

catalogue for improvement, we identified a disubstituted 2-aminoethyl-quinazoline, "3B1", with a moderate binding affinity and high selectivity for the higher-order hTERT G4 ($K_d =$ 3 μM). Extensive docking and MD studies with *post hoc* free energy calculations indicate that this molecule binds across inter-G4 junctional regions specific to the higher-order quadruplex. We further show that 3B1 reduces hTERT mRNA and protein levels in breast cancer cells, positioning it as a strong candidate for further refinement as a selective telomerase inhibiting drug.

## Materials and methods

### Oligonucleotides

Oligos (Table 1) were purchased from either IDT (Integrated DNA Technologies, Coralville, IA) or Sigma-Aldrich (St. Louis, MO) with standard desalting. FRET-labeled oligos used in FTSA experiments had 6-FAM (6-carboxyfluorescein) attached to their 3' end and TAMRA (5-carboxytetramethylrhodamine) attached to their 5' end. Upon receipt, stock oligos were dissolved in MilliQ ultrapure water (18.2 MΩ x cm at 25˚C) at concentrations between 0.1 and 1 mM and stored at -20.0˚C until use. Folding was achieved by diluting stock oligos into their respective buffer and heating to 99.9˚C in a water bath for 20 minutes, followed by slow cooling overnight and subsequent storage at 4˚C until use. Unlabeled oligonucleotide concentrations were determined from their extinction coefficients ($\varepsilon_{260}$) using the nearest-neighbor method in alkaline buffer (pH 11.5) [51]. FRET-labeled oligo concentrations were determined using the extinction coefficient of 6-FAM ($\varepsilon_{495} = 75,000$ cm$^{-1}$ M$^{-1}$).

### Chemicals

All small molecules were purchased as 1 or 5 mg quantities from the distributer Molport.com, diluted to 10 mM in DMSO, and stored at -80˚C until use. Compounds 3B and 3B1 were re-purchased as needed (3B from ChemBridge.com and 3B1 from Molport.com). Compound concentrations were determined using molar extinction coefficients determined from UV-Visible spectroscopy measurements of stock compounds diluted into potassium phosphate buffer at pH 7.2.

**Table 1. Oligonucleotides used throughout this study.**

| Name | Oligonucleotide Sequence 5' to 3' | Length | MW | ε260 (M$^{-1}$cm$^{-1}$) |
|---|---|---|---|---|
| TERT-FL | GGGGAGGGGCTGGGGAGGGCCCGGAGGGGGCTGGGCCGGGGACCCGGGAGGGGTCGGGACGGGGCGGGG | 68 | 21633 | 672671 |
| PQS1 | AGGGGAGGGGCTGGGGAGGGC | 20 | 6369 | 202900 |
| PQS12 | AGGGGAGGGGCTGGGGAGGGCCCGGAGGGGGCTGGGCCGGGGACCCGGGA | 49 | 15523 | 478700 |
| PQS23 | AGGGGGCTGGGCCGGGGACCCGGGAGGGGTCGGGACGGGGCGGGG | 45 | 14278 | 436500 |
| Ctrl HP | GCATGCTTTTGCATGC | 16 | 4863 | 141300 |
| AT duplex | ATATATATATCCCCATATATATAT | 24 | 7269 | 205700 |
| GC duplex | GCGCGCGCGCTTTTGCGCGCGCGC | 24 | 7339 | 166300 |
| c-Myc G4 | TGAGGGTGGGTAGGGTGGGTAA | 22 | 6992 | 228700 |
| H-Tel G4 | AGGGTTAGGGTTAGGGTTAGGG | 22 | 6967 | 228500 |
| GC Triplex | CCCCCCCCCTTTTGGGGGGGGGTTTTCCCCCCCC | 32 | 9632 | 261600 |
| DNA/RNA duplex | GGGTTAGGGTTTTrCrCrCrCrUrArArCrCrC | 22 | 6785 | 202900 |

## Buffers

All buffer reagents, unless otherwise specified, were purchased from Sigma-Aldrich. BPEK buffer (8 mM sodium phosphate, 1 mM EDTA, pH 7.2) was used throughout supplementation of KCl (25, 100, 185 mM) as indicated. For cell studies: RIPA buffer (20 mM Tris-HCl, pH 7.4, 150 mM NaCl, 1 mM $Na_2$EDTA, 1% Triton X-100, 1% sodium deoxycholate, 0.1% SDS, 1 mM PMSF, Roche protease inhibitor cocktail [Sigma cat# 11697498001]), Towbin buffer (25 mM Tris, 192 mM glycine, 10% MeOH, pH 8.3), TBST (20 mM Tris, 150 mM NaCl, 2.7 mM KCl, 0.1% Tween-20, pH 7.4). TBST was autoclaved and all buffers were filtered through 0.2 µm filter paper prior to use.

## Preparative size exclusion chromatography (SEC)

Oligonucleotide purification was done using SEC as detailed previously [52]. Briefly, oligonucleotides were annealed at concentrations of 40–100 µM in their respective buffers, filtered through 0.2 µm filters, and injected onto an equilibrated Superdex 75 16/600 SEC column (GE Healthcare 28-9893-33) using a Waters 600 HPLC system. The flow rate was maintained at 0.5 mL/min and sample fractions were collected every 2 minutes from 100 to 180 minutes run time. The molecular weights of fractionated species were estimated based on a regression analysis of elution time vs. log(MW) of protein standards (Sigma #69385), with elution profiles monitored at 260 nm for DNA and 280 nm for protein. Purifications were carried out at room temperature and fractionated samples were stored at 4˚C prior to concentration and downstream analysis.

## *In silico* drug screening

Virtual screening was performed using Surflex-Dock 2.601 [53] on the KY Dataseam computing grid (http://www.kydataseam.com/) using over 40 million compounds from the ZINC 2014 and 2016 drug-like libraries, containing 24,877,119, and 17,244,856 compounds, respectively [54]. The all-parallel G-quadruplex hTERT model created previously [31, 33] was used as the receptor. Twelve docking sites were chosen by targeting the G4-G4 junctions, loops, and grooves. Docking in Surflex-Dock was carried out as previously described [55]. Virtual screens were preformed using the 2014 and 2016 libraries at each site (resulting in 24 separate screens), and the top scoring 500 poses were analyzed by hierarchical clustering in Schrodinger's Canvas [56] application using each molecule's binary fingerprint and Tanimoto similarity criteria. The highest ranking 100 centroid molecules (i.e., most representative scaffolds of each respective clade) were subsequently inspected visually for unique molecular structures and in total 69 of these were available for purchase.

## Fluorescence thermal shift assays (FTSA)

Small molecule screening by FTSA was performed in 96-well plates on an Applied Biosystems StepOnePlus Real-Time PCR instrument as adapted from previous work [57]. Briefly, 10 mM compound stock solutions in DMSO were used to create 96-well stock solution plates by diluting each compound to 2x final concentration in potassium phosphate buffer. The same volume of DMSO was used as control. FRET-labeled oligos were quantified after annealing by UV-Vis and diluted to 2x final concentration. FTSA reaction mixes were made up in 96-well Applied Biosystems MicroAmp PCR plates by mixing 10 µL of 2x compound solution (or buffer/DMSO control) with 2x FRET-labeled DNA to yield 20 µL of 1x reaction mix. For screening, a final ratio of 200:1 [compound]:[DNA] was used ([100 µM compound]:[0.5 µM DNA]). Plates were subsequently spun down at 1250 rpm for 2–3 minutes in a benchtop centrifuge to remove

bubbles. Samples were denatured by ramping the temperature from 20.0˚C to 99.8˚C in 0.2˚C increments at a rate of approximately 0.7˚C/min. Fluorescence quenching of 6-FAM was monitored at each 0.2˚C step using the instrument's onboard FAM filter over the entire reaction, providing a melting curve. Melting temperatures ($T_m$) were determined from the first derivative of the normalized melting curves [58], and differences in melting temperatures ($\Delta T_m$) were determined by taking the difference of control and sample wells:

$$\Delta T_m = T_{m,sample} - T_{m,control}$$

where $T_{m,sample}$ and $T_{m,control}$ are the melting temperatures of the sample and control, respectively. Measurements are averages of triplicate wells from 3 measurements made on independent days unless otherwise specified.

## Competition dialysis

Competition dialysis assays were conducted as described [59] in a 96-well plate format with minor changes. Nucleic acids were annealed in a 185 mM K+ phosphate buffer, SEC purified, and concentrated to create stock solutions of 20–100 μM (strand concentration). From these stocks, nucleic acids were diluted to 75 μM working concentration based on monomeric unit, where each nucleotide, base pair, triplet, or tetrad is considered a single monomeric unit (e.g. the full-length hTERT has 9 contiguous tetrads, or 9 monomeric units, whereas c-Myc only has 3 tetrads, so hTERT would be at 1/3 the final strand concentration of the c-Myc G4). Into each dialysis membrane was added 200 μL of nucleic acid sample or buffer only (control). The dialysis membranes were then submerged in at least 2 or 5 mL of a 5 μM solution of compound and allowed to incubate on a rocker overnight at RT. After 24 hours, 180 μL of sample was removed and added to a UV-permissible 96-well plate, along with 20 μL of a 10% Triton X-100 solution to disrupt ligand interactions with receptors. The ligand concentration was then determined by measuring the ligand absorbance using a Tecan Safire II plate reader (Tecan, Männedorf, Switzerland). Total concentration of bound compound ($C_b$) was determined as follows:

$$C_b = C_t - C_f$$

Where $C_t$ is the total concentration of ligand in the sample membrane and $C_f$ is the concentration of compound in the buffer only membrane (which did not deviate from the 5 μM in the dialysis buffer). Calculation of apparent binding affinities ($K_{app}$) was achieved using the following equation:

$$K_{app} = C_b / \left\{ C_f \times \left( [NA]_{total} - C_b \right) \right\}$$

Where $[NA]_{total}$ is the total DNA concentration in the well (0.000075 M).

## Circular dichroism spectroscopy (CD)

CD melting studies and spectra were collected on a Jasco-710 spectropolarimeter (Jasco Inc. Eason, MD) equipped with a Peltier thermostat regulated cell holder and magnetic stirrer. CD and melting spectra were collected using the following instrument parameters: 0.5 or 1 cm path length quartz cuvette, 210 or 240 to 340 nm wavelength range, 1.0 nm step size, 200 nm/min scan rate, 1.0 nm bandwidth, 2 s integration time, and 4 scan accumulations. Spectra were recorded at 20.0˚C and melting spectra were collected over a range of 4˚C to 98˚C with 4˚C step intervals, 4˚C/min ramp speed, and a 1-minute equilibration time at each temperature before acquisition. Spectra were corrected by subtracting a matched buffer blank. Spectra were

normalized to molar circular dichroism ($\Delta\varepsilon$) based on DNA strand concentration using the following equation:

$$\Delta\varepsilon = \theta/(32982cl)$$

where $\theta$ is ellipticity in millidegrees, $c$ is molar DNA concentration in mol/L, and $l$ is the path length of the cell in cm. For melting experiments, the concentration of DNA was 1.1 μM and compounds were 25 μM (except for the positive control, BRACO-19, which was 2.5 μM). The same volume of DMSO was used in the control melts. Melts were performed in duplicate on separate days.

## Microscale thermophoresis (MST)

MST traces were generated with a Monolith Nanotemper instrument using its MO.Control v2.0.4 software. Optimal buffer conditions were determined to be BPEK buffer at pH 7.2 supplemented with 25 mM KCl. Titration solutions consisted of 20 nM Cy5-labled (and SEC purified) hTERT-FL mixed in a 16-point serial dilution series of compounds. Concentration series were made up with a high concentration of 200 μM ligand and 1:1 dilution down to ~3 nM with a final constant concentration of 1.5% DMSO. Dilutions were made in 384 well plates at a volume of 20 μL and incubated in the dark for at least 5 minutes before beginning MST analysis. For 20 nM of Cy5-labeled DNA, MST excitation was consistently between 20–30% on medium power. Data analysis was performed in the program PALMIST [60, 61] using the cold fluorescence mode and a 1-site binding model.

## Post screening molecular dynamics simulations and free energy calculations

Starting coordinates for small molecule-G4 complexes were based on the output of flexible docking performed using Glide XP [62] with the Maestro [63] suite using the hTERT-FL model created previously [31, 33] and shown in Fig 1. Briefly, Glide's receptor grid generation tool was used to create docking sites corresponding to residues used in generating the original protomols from the Surflex-Dock screen. Next, the LigPrep module was used to optimize each of the 3B derivatives (3B, 3B1-5). Each molecule was protonated at pH 7.2 (±0.2) with Epik [64] and all tautomers generated were retained. The tautomeric forms of each derivative were then used in Glide XP flexible docking. The highest scoring (based on Glide score) tautomer and conformation of each derivative at each docking site were then subjected to 5 ns molecular dynamics simulations and free energy calculations. The MD and free energy calculations were performed as follows using an in-house script, Docking Free Energy Calculator (DFEC), described previously [65]. The PDB structures were imported into the xleap module of AMBER18 [66] with protein and DNA forcefields ff14SB and OL15, with the radii set mbondi2, neutralized with K+ ions, and solvated in a rectangular box of TIP3P water molecules with a 12 Å buffer distance. Small molecules were parameterized using the Antechamber [67] package with general AMBER force field (GAFF) [68] and AM1-BCC atomic charges [69]. All simulations were equilibrated using sander and brought to 300 K and 1 atm using the following steps: (1) minimization of water and ions with weak restraints of 10.0 kcal/mol/Å on all nucleic acid and ligand residues (2000 cycles of minimization, 500 steepest decent before switching to conjugate gradient) and 10.0 Å cutoff, (2) heating from 0 K to 100 K over 20 ps with 50 kcal/mol/Å restraints on all nucleic acid and ligand residues, (3) minimization of entire system without restraints (2500 cycles, 1000 steepest decent before switching to conjugate gradient) with 10 Å cutoff, (4) heating from 100 K to 300 K over 20 ps with weak restraints of 10.0 kcal/mol/Å on all nucleic acid residues, and (5) equilibration at 1 atm for 100 ps with

weak restraints of 10.0 kcal/mol/Å on nucleic acids. The output from equilibration was then used as the input for the 5 ns of production MD simulations using pmemd with GPU (CUDA) acceleration in the isothermal isobaric ensemble (P = 1 atm, T = 300 K). Periodic boundary conditions and PME were used. 2.0 fs time steps were used with bonds involving hydrogen frozen using SHAKE (ntc = 2). Trajectories were analyzed using the CPPTRAJ module in the AmberTools18 package. Calculations of theoretical relative Gibb's free energy (ΔG) of ligand-receptor complexes was achieved using the single-trajectory MMPBSA method [70].

## Cell culture

MCF7 breast cancer cells (ATCC HTB-22) were maintained in 5% $CO_2$ at 37°C and 95% humidity in EMEM media supplemented with 10% heat-inactivated FBS, penicillin, and streptomycin. Cells were treated with compounds at the indicated times and concentration by mixing the compounds directly into the EMEM media, washing the cells of old media, and replacing with freshly treated media every 24 hours.

## Quantitative reverse-transcription and polymerase chain reaction (RT-PCR)

MCF7 cells were seeded at $2 \times 10^5$ cells/well in 6-well plates. After overnight attachment, media was replaced and treated with compounds or DMSO alone (control), followed by 2 minutes of gentle mixing before placing back in the incubator. Media replacement and compound treatment was repeated twice more in 24-hour intervals such that the end time point was 72 hours of total treatment time. After 72 hours, cells were aspirated and washed before harvesting of total RNA with PureLink RNA mini kit (Invitrogen, #12183018A), followed by reverse transcription into cDNA using a high-capacity reverse transcription kit (Applied Biosystems #4368813). Quantitative PCR was performed using a standard SYBR Green Master Mix (Applied Biosystems #4309155) in 96-well plates on an Applied Biosystems StepOnePlus RT-PCR system using the ΔΔCt method. Primers were from PrimerBank (https://pga.mgh.harvard.edu/primerbank/) and verified as specific based on monophasic transitions during thermal denaturation. Primers (5' to 3'): hTERT F-TCCACTCCCCACATAGGAATAGTC, R-TCCTTCTCAGGGTCTCCACCT, c-MYC F-CGTCTCCACACATCAGCACAA, R-ACTGTCCAACTTGACCCTCTTG, GAPDH F-TGCACCACCAACTGCTTAGC, R-GGCATGGACTGTGGTCATGAG.

## Immunoblotting

Total protein extracts prepared from MCF7 cells treated with either sterile $dH_2O$, DMSO, or compound 3B1 were subjected to SDS-PAGE followed by wet transfer to nitrocellulose membranes. After 72 hr treatment cells were washed with ice-cold PBS 3x before lysing in ice-cold RIPA buffer. Cell supernatants were collected after centrifugation at 12,000 rpm for 5 minutes and subsequently stored at -80°C until use. Protein quantification was achieved using a Pierce BCA protein quantification kit (ThermoFisher cat# 23225). Proteins were resolved using an SDS-PAGE gel procedure with 4–15% Mini-PROTEAN TGX Precast gels (Bio-rad cat# 4561086). Transfer to nitrocellulose membranes was done over 4–6 hours at 4°C in a Towbin buffer with 10% MeOH at pH 8.3 and a constant 15 V. Blots were rinsed 3x with TBST, blocked with TBST supplemented with 5% milk, and subsequently incubated with primary anti-hTERT antibody (Abcam, ab230527–1:500 dilution) or GAPDH (Abcam, ab9485–1:2,000 dilution) at 4°C overnight. Visualization was achieved by incubation and visualization of an anti-rabbit Alexa Fluor 488 conjugated secondary antibody (Abcam cat# 150077) using a PharosFX imaging system.

### Data analysis

All data fitting, statistical analysis, and graphing was performed using Origin 2020 (OriginLab Corporation, Northampton, MA, USA).

### Molecular visualizations

All molecular visualizations of MD trajectories and models were performed in UCSF Chimera v1.12 [67] and Maestro [63].

## Results

### FTSA screening and enrichment for hTERT G4 stabilizing molecules

Initial *in vitro* testing of the 69 small molecules from virtual screening was performed using a fluorescence thermal shift assay (FTSA) (described elsewhere [50]). Due to inherent difficulties in the synthesis of long G-rich nucleic acids, the full-length 68-nt long hTERT core promoter sequence ("hTERT-FL") (Fig 1A) could not be synthesized with FRET-pair labels. Instead, initial screening was performed with FRET-labeled fragmented hTERT sequences, namely PQS1, PQS1 & PQS2 ("PQS12"), and PQS2 & PQS3 ("PQS23") (Fig 1B and Table 1). PQS1 has previously been structurally characterized by NMR [35], and PQS12 and PQS23 have been characterized by integrative structural approaches [31]. Each sequence adopts a parallel conformation in potassium-containing buffer (S1 Fig). A control hairpin, "Ctrl HP", was included throughout the FTSA studies. This control serves two purposes; first, it's an initial indicator of non-specific interaction with a duplex DNA and, second, it serves as a control for non-specific interactions with the FRET labels.

Fig 2A is a heatmap of $T_m$ shift vs. hTERT G4 or hairpin control sequence. Of the 69 compounds, only 33 (shown) had stabilizing or destabilizing effects on the hTERT G4s. The top three stabilizing compounds were designated "1, 2, and 3" and had $T_m$ shifts of 0.5–3.3˚C. Initial screening was performed in the presence of high potassium (185 mM K$^+$), as this is the potassium concentration in which the sequences had previously been structurally characterized. However, it is well established that G4 stability depends on K$^+$ concentration [71], and so the resulting $T_m$ shifts were modest and, in some cases, within the noise of the measurement. Therefore, to confirm that compounds 1–3 stabilized the PQS12 and PQS23 sequences, subsequent measurements were made at lower potassium concentrations (100 mM K$^+$). Fig 2B shows that compound 1 stabilizes PQS23 and has a negligible effect on PQS12 and Ctrl HP. Conversely, compounds 2 and 3 significantly stabilize PQS12, both with $\Delta T_m$s $\approx$ 14–15˚C. However, compound 3 destabilized PQS23 and, to a lesser extent, the Ctrl HP. The chemical structures and ZINC IDs for compounds 1–3 are shown in Fig 2C.

To our knowledge compounds 1–3 have not been reported as G-quadruplex DNA scaffolds. To increase our chances of identifying tightly binding and selective hTERT G4 binders, we next used a catalogue-based structure activity relationship ("SAR by catalogue") approach, whereby we searched the chemical distributor Molport.com for molecules with high (>90%) similarity to compounds 1–3. In total, 25 more molecules were purchased. Each new molecule has an ID with letter identifiers appended after their first-generation ID number. Fig 2D–2F shows the FTSA screening results of the second-generation compounds. There were six derivatives of compounds 2 and 3 that exhibited a greater ability to stabilize the PQS12 and PQS23 quadruplexes (compounds 2R, 2S, 2T, 3A, 3B, and 3Y—see S2 Fig for molecular relationships to 1$^{st}$ generation). Among these six, only compound 3A appeared to affect the $T_m$ of the hairpin control. In contrast, there was only one derivative of compound 1, compound 1N, with increased stabilization (although it destabilized PQS12, Fig 2F). Compounds 1N, 2R, 2S, 3A,

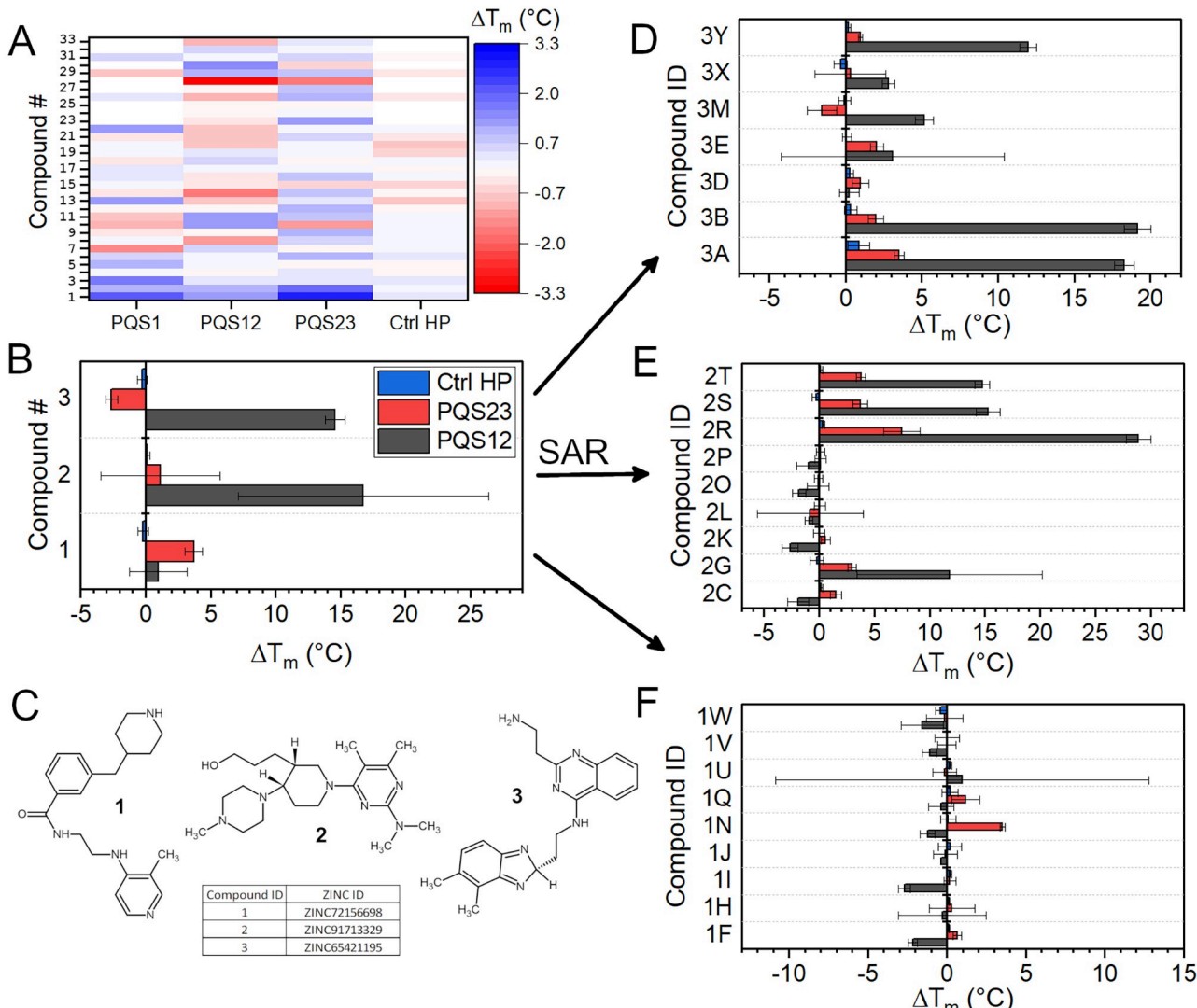

**Fig 2. FTSA drug screening and "Catalogue" SAR.** (A) Heatmap of $\Delta T_m$s for 33 of the original 69 compounds identified in virtual screens plotted against the given hTERT sequences (numbering is arbitrary, compounds not shown had no apparent effect on $T_m$, and were not further pursued). The initial screen was performed at a potassium concentration of 185 mM $K^+$. (B) Re-analysis of compounds 1–3 from A at a lower potassium concentration of 100 mM $K^+$. (C) Structures and ZINC IDs of compounds 1–3. (D-F) FTSA analyses of the SAR compounds derived from 1–3. All FTSA results are given as averages of triplicate measurements from three independent days. Whiskers represent the standard deviation about the mean.

and 3B were all confirmed by circular dichroism melting experiments to bind and stabilize hTERT-FL (S3 Fig).

## Selectivity assessment and SAR enhancement of 2nd generation compounds

We next investigated the selectivity of the second-generation compounds by competition dialysis, as it is ideal for rapid and direct comparisons of selectivity to multiple alternative DNA topologies [72, 73]. Fig 3 displays the results of the competition dialysis experiment. Only compounds 3A, 3B, and 2S had UV-visible spectra convenient for this type of experiment. The AT and GC duplexes are typical B-form DNA, representing the most prominent off-target in the cell. A GC triplex and two forms of G-quadruplex, a mixed hybrid, H-Tel, and parallel, c-Myc,

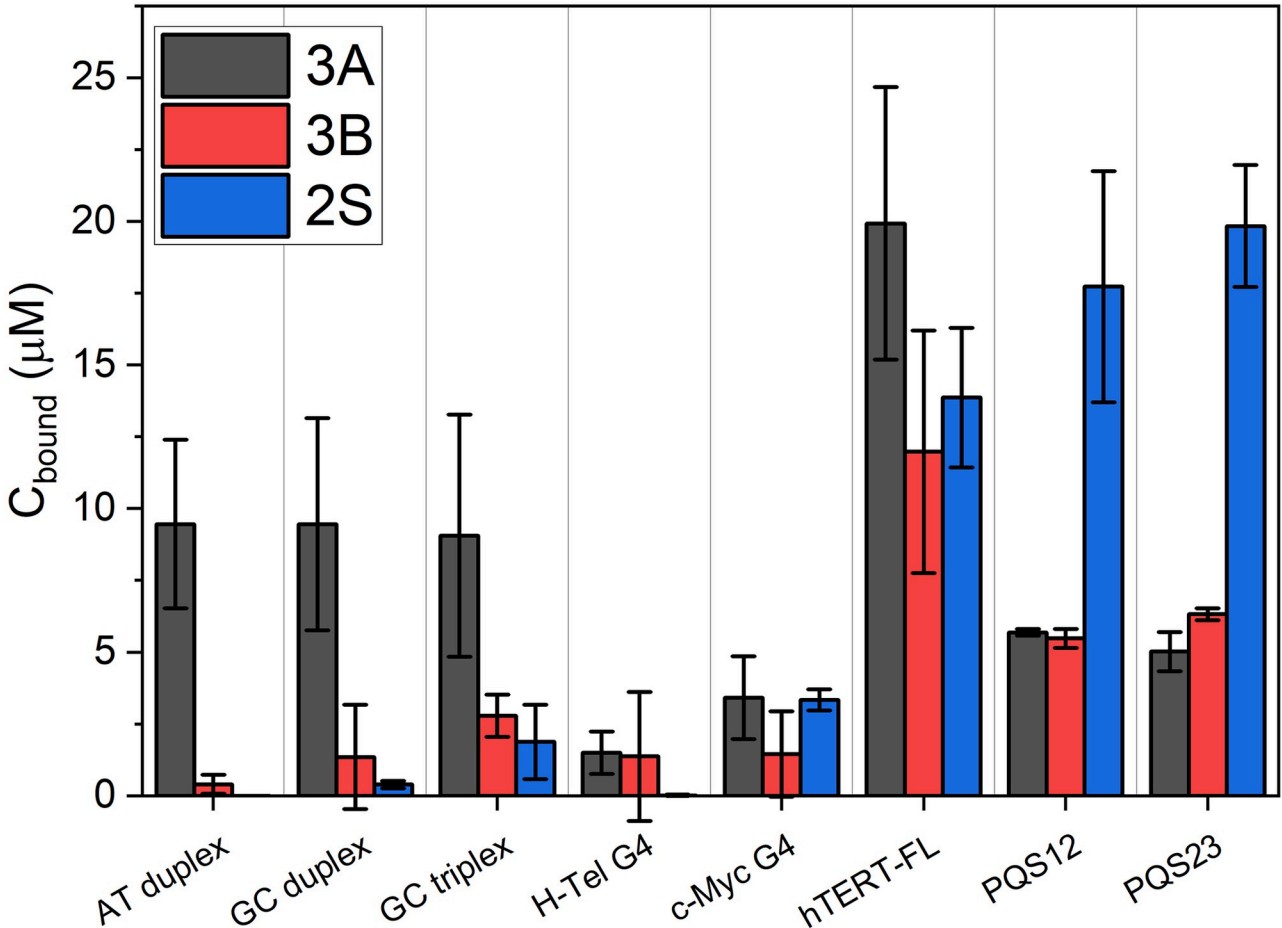

**Fig 3. Competition dialysis results.** The bound concentration ($C_{bound}$) is the concentration of compound in the dialysis well less that of the "free" compound concentration in the dialysate. Bars and whiskers represent average and standard deviation about the mean from the average of duplicate measurements made on four independent days.

were included as non-B off-target topologies. Compound 3A displays significant binding to duplex DNA, to triplex DNA and to the two G-quadruplex DNA topologies, showing that it has poor selectivity. Conversely, compounds 3B and 2S exhibited selectivity for TERT-FL over monomeric G4s, duplex, and triplex DNAs. On average, hTERT-FL had nearly double the amount of 3B locally enriched relative to its truncated fragments alone, whereas the opposite is observed for compound 2S. It should be noted that the *monomeric unit concentrations* used in the competition assay are the same (i.e. base pairs for duplex forms, triplets for triplex, and tetrads for quadruplexes). This effectively allows for direct comparisons in affinity across all receptors, irrespective of size, and accounts for multiple binding sites [59]. The apparent dissociation constants, $K_d$, are 24 µM and 22 µM for 3B and 2S with respect to TERT-FL and >140 µM with respect to the duplex, triplex, and H-Tel and c-Myc G4s. Overall, these data show that 3B and 2S have a strong preference for G-quadruplex DNA over duplex and triplex conformations and have a strong preference for the higher-order hTERT G4s over other topologies.

Since compound 3B showed good selectivity for hTERT-FL over other forms of DNA, including the truncated forms PQS12 and PQS23, we next used it in a third round of SAR by

catalogue to obtain a third generation of derivatives based on its scaffold (Fig 4). These derivatives were designated as 3B1-5. The FTSA results in Fig 4A reveal that 3B1 and 3B2 have greatly improved thermal stabilizing effects on PQS23 with no discernable interaction with the control hairpin. Compounds 3B3 and 3B4 had reduced thermal stabilization, suggesting that the ethylamine moiety plays an important role in binding. Consistently, 3B5 had greatly reduced thermal stabilization even though it retained the core structure common to both 3B and 3B1.

The FTSA assay is excellent as a rapid indicator of compound interaction. However, as we've recently shown [65], $T_m$ shifts do not necessarily correlate with binding affinities. Therefore, we next screened each of the 3B derivatives using microscale thermophoresis (MST) against a Cy5-labeled full-length hTERT G4. The measured dissociation constants, $K_d$, are inset next to each molecular structure in Fig 4B, and representative binding curves are given in S4 Fig. Consistent with the FTSA data, all but compound 3B4 bound hTERT-FL. Among the five that bound, 3B1 showed the highest affinity ($K_d$ = 3 μM, quadruplicate measurements). Regression analyses of the measured $K_d$ values *versus* $\Delta T_m$s emphasizes that $T_m$ shifts are not necessarily correlated with binding affinity (S5 Fig—Pearson's $r^2$ = 0.190 and 0.028 for PQS12 and PQS23, respectively) as it is a function of multiple thermodynamic parameters [65].

## Exploration of 3B1 binding interactions

To understand how 3B1 and other derivatives might stabilize the hTERT-FL quadruplex, we conducted extensive post-screening re-docking and MD simulations (Fig 5). Docking sites around the loops and grooves of the hTERT model were generated based on the residues used in creating the original 12 protomols from Surflex-Dock screening, followed by Glide XP [62] flexible docking at each site. In total, five Glide grid sites encompassing the original 12 protomols were generated. The Glide docking sites were named for the residue most representative of the docking site (residues DG4, DG22, DG38, DG39, and DG55). Fig 5A and Table 2 show the calculated binding free energy values (if applicable) for the receptor/tautomer pair with the highest predicted affinity at each docked site and for each compound (i.e. multiple 3B1 tautomers and conformers were docked and simulated at site DG22, however, only the most stable predicted complex is reported). In total, 425 ns of explicit solvent MD simulations were carried out.

Fig 5A shows that compounds 3B, 3B1, and 3B2 have a distinct preference for sites DG22 and DG38 over the other groove sites identified. As anticipated from FTSA, these two sites are situated at the junctions between PQS1 & PQS2 and PQS2 & PQS3, respectively, consistent with their ability to stabilize both sequences in the FTSA assays. Fig 5B–5E highlight 3B1's interaction across these two junctions. Fig 5B & 5C are 2D schematic representations of the residues that 3B1 hydrogen bonds with across the two sites, illustrating how the binding sites are formed among the higher-order organization and not of a single G4 domain. In both cases the positively charged ethylamine group is predicted to bind and stabilize across the backbone phosphate groups spanning the G4-junctional regions (indicated by the purple dashed lines in Fig 5B and 5C and the yellow lines indicating H-bonds in Fig 5D and 5E). At site DG22 3B1 is predicted to have multiple H-bonds to waters that bridge residues associated with the binding site with its hydrophobic ring systems projecting towards the hydrophobic G-tetrad core pocket (see also S6 Fig). Conversely, at site DG38, 3B1 is not buried in a pocket, but rather projects its ether group away from the quadruplex into the bulk water while stacking onto an adjacent cytosine with its quinazoline ring system (see Fig 5E & S7 Fig).

We subsequently used the program SiteMap [74] to quantitatively assess the degree of "druggability" of each of the two 3B1 binding sites. SiteMap druggability scores (Dscores)

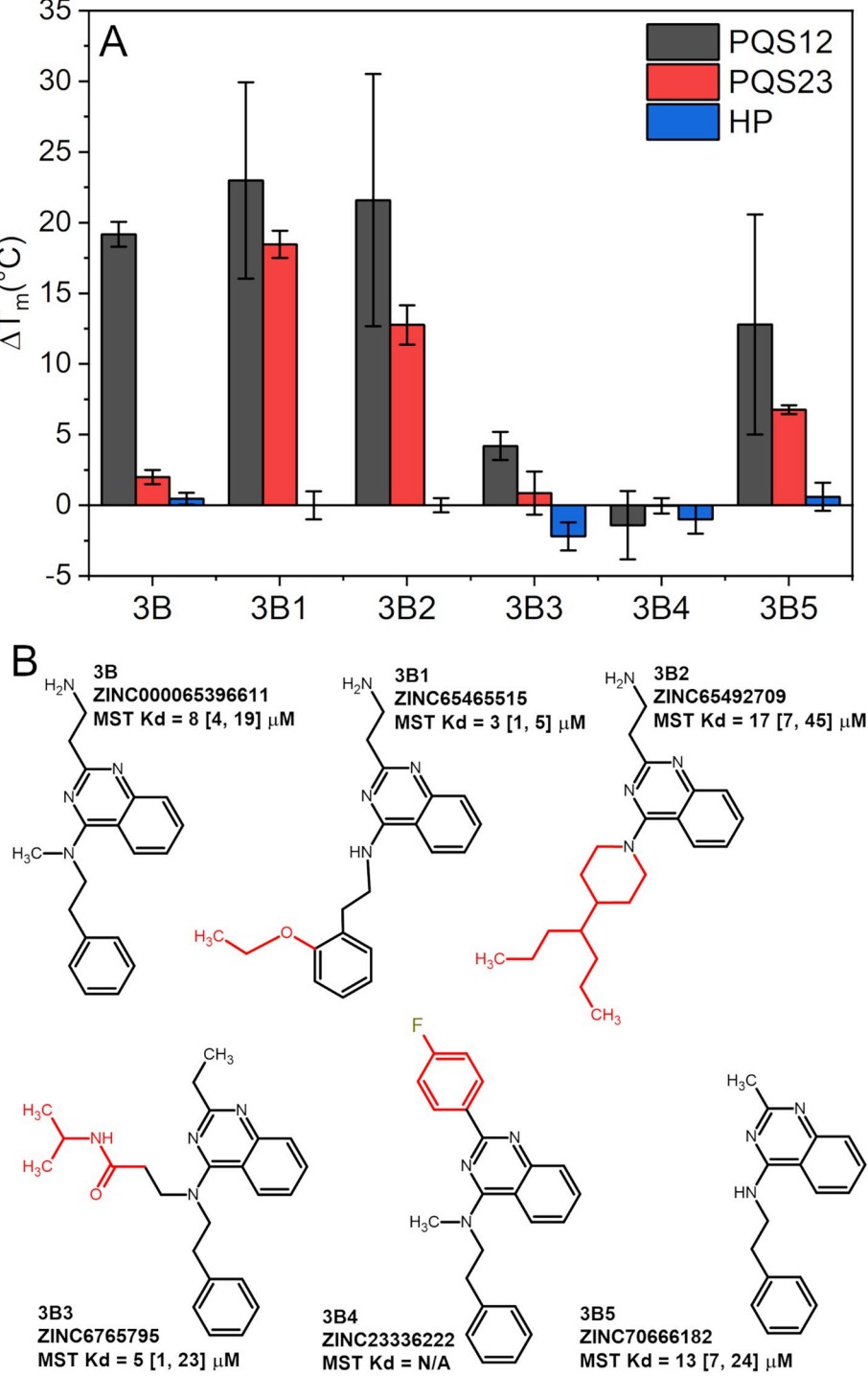

**Fig 4. FTSA and MST SAR by catalogue results for 3B and its derivatives.** (A) FTSA results of each derivative with their respective targets shown as average and standard deviation from triplicate measurements made on three independent days. (B) Molecular structure, ZINC ID, altered substituents (shown in red), and MST dissociation constant values ($K_d$) with 68.3% confidence intervals in brackets. "N/A" indicates that no binding was observed. See S4 Fig for representative curves and fits.

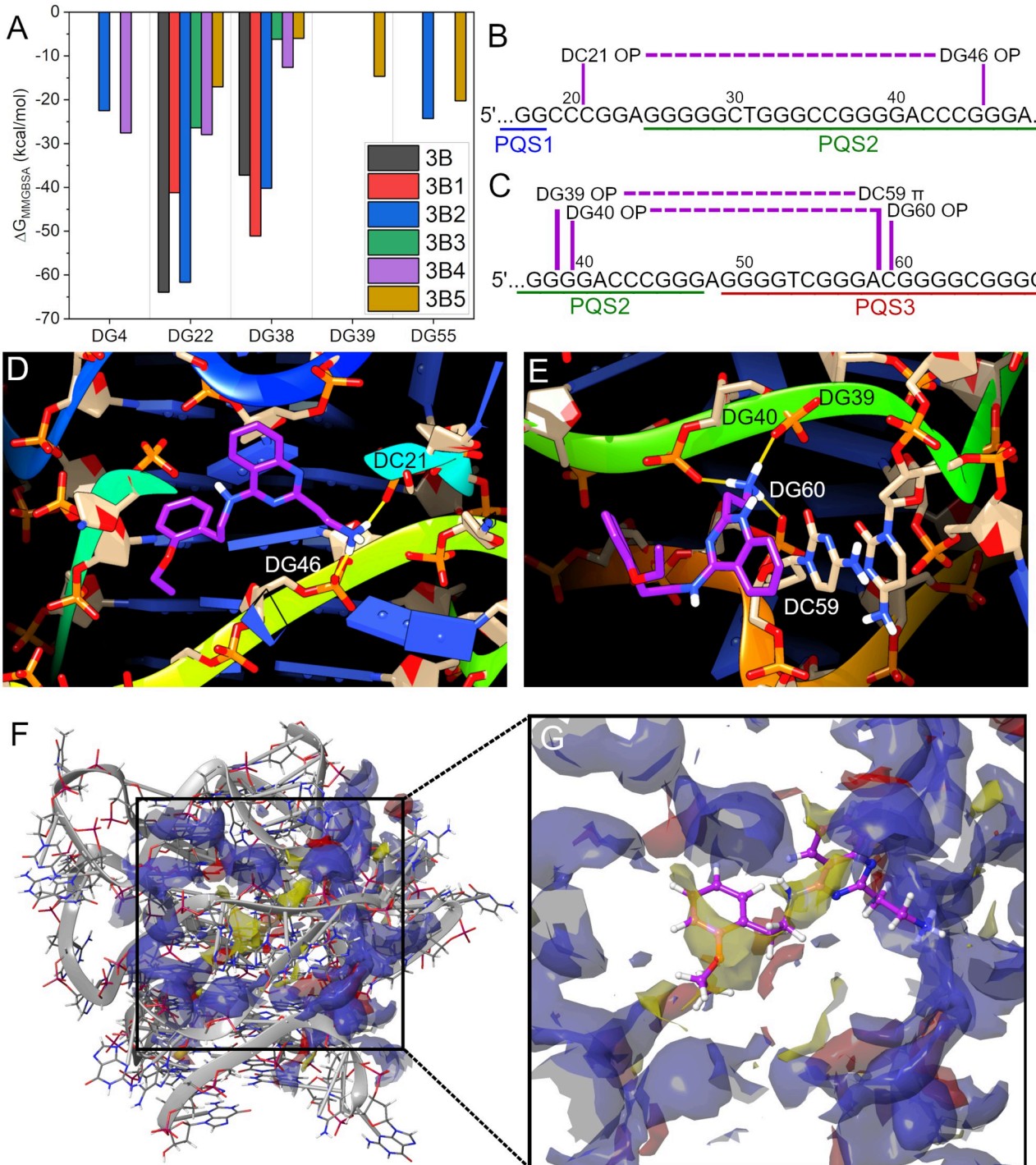

**Fig 5. Glide XP docking and post-MD interactions of 3B1 with the higher-order hTERT G4.** (A) Summary of free energies calculated for each derivative at each docking site. The site names "DG#" are used to distinguish where on the quadruplex the compounds were docked. Values are theoretical free energies of binding using the implicit GBSA model calculated over 5 ns of production simulation in explicit solvent. (B, C) Schematic representations of the regions of the hTERT G4 sequence where 3B1 is predicted to bind. The purple lines and dashes are used to illustrate that in docking site "DG22" (D), the 3B1 binding site spans PQS1 and PQS2 regions, and at site "DG38" (E), 3B1 spans PQS2 and PQS3 via phosphate backbone and stacking interactions. (F) The hTERT higher-order G4 with DG22 SiteMap region overlaid. The phosphate backbone is shown as gray ribbons with PQS1 (the 5' end) at the top. Ligand donor, ligand acceptor, and hydrophobic regions are shown as blue, red, and yellow, respectively. (G) The SiteMap region blown up with 3B1 overlaid in purple as ball and stick representation. The hydrophobic regions of 3B1 are situated within the predicted hydrophobic pocket (yellow), with its hydrogen donating nitrogens near or within the predicted ligand donor regions and its ether oxygen near a ligand acceptor region.

**Table 2. Tabulated free energy results from 5 ns MD simulation.** Free energies, if available, are given for each compound at each docking site. The corresponding compound RMSD is also given as an indication of change from initial docked conformation at the start of simulation. The values reported are for the largest negative free energy value for each ligand/conformer/tautomer at each site. The parenthesis next to docking site indicates the total number of simulations at each site (i.e., 37 different combinations of derivatives with favorable docking scores were simulated at site DG22). Instances of missing values indicates that either there was no docking solution or that the docking score indicated an unfavorable interaction.

| Site | DG4 (6) | | DG22 (37) | | DG38 (34) | | DG39 (2) | | DG55 (6) | |
|---|---|---|---|---|---|---|---|---|---|---|
| Compound ID | ΔG | RMSD | ΔG | RMSD | ΔG | RMSD | ΔG | RMSD | ΔG | RMSD |
| **3B** | - | - | -63.9 | 1.6 | -37.2 | 2.0 | - | - | - | - |
| **3B1** | - | - | -41.2 | 0.9 | -51.1 | 2.6 | - | - | - | - |
| **3B2** | - | - | -61.7 | 1.2 | -40.2 | 1.4 | - | - | -24.3 | 2.3 |
| **3B3** | -22.5 | 1.7 | -26.4 | 1.6 | -6.2 | 2.8 | - | - | - | - |
| **3B4** | - | - | -28.0 | 1.0 | -12.6 | 2.8 | - | - | - | - |
| **3B5** | -27.6 | 1.4 | -17.0 | 0.8 | -6.0 | 2.0 | -14.7 | 2.9 | -20.2 | 1.9 |

range from 0.63 (undruggable) to 1.11 (druggable), with anything under ~0.87 considered "difficult" [75]. The Dscores for sites DG22 and DG38 were 0.93 and 0.80, respectively, indicating that site DG22 is druggable and DG38 is difficult to target. Fig 5F and 5G show the Site-Map for DG22 overlaid on hTERT and 3B1, respectively. SiteMap volumes are colored based on if the region is predicted to be a ligand donor (blue), ligand receptor (red), or hydrophobic pocket (yellow). 3B1 is situated with its mostly hydrophobic ring systems within the hydrophobic pocket, with its nitrogen H-bond donors in the ligand donor regions. The ethoxy moiety is adjacent to an H-bond acceptor region (red region to the lower left of the ethoxy in Fig 5G) that may be occupied by the bridging waters predicted from simulation (S6 Fig). Taken together with the FTSA and MST analyses, this provides atomistic rationalization for the role of the 2-aminoethyl substituted quinazoline group in stabilizing the inter-G4 PQS12 region of hTERT-FL. The 3B1 core scaffold fits snugly within a hydrophobic cavity formed in the groove spanning PQS1 and PQS2 and has predicted water-facilitated interactions that may be important in binding, such as the bridging water H-bond interactions from the nitrogen at the 4-position of the quinazoline group (compare PQS23 stabilization of 3B1, 3B2, and 3B5 with 3B, 3B3, and 3B4 in Fig 4).

## Biological assessment of compound 3B1

We next wanted to know if 3B1 could induce a biological response in cells with a wild-type core promoter and elevated hTERT expression [76]. We began with quantitative PCR studies in MCF7 breast cancer cells using 10 μM compound and 72-hour treatment (Fig 6). The concentration and duration of treatment were determined based on cell viability studies (not shown) and prior work by Kang *et al.* [38] who determined that 72 hr. treatment was appropriate for achieving a reduction in hTERT mRNA levels by the ligand GTC365. Treatment of MCF7 cells with compounds 3B1 and GTC365 showed a significant reduction in hTERT mRNA relative to DMSO treated control cells. As a test for selectively, we also measured c-Myc mRNA levels. c-Myc a gene that it is established can be modulated by G-quadruplex stabilization and can act on hTERT's promoter via E-box binding leading to changes in its expression. Neither compound influenced c-Myc levels. To confirm that hTERT protein levels were also affected by 3B1 treatment, we conducted western blotting of MCF7 cells, which showed a dose-dependent reduction in protein levels with $IC_{50}$ ~ 1–5 μM (Fig 6B). Collectively, these studies provide strong evidence for 3B1 as a selective, moderate affinity, drug-like small molecule that targets the higher-order hTERT core promoter G4.

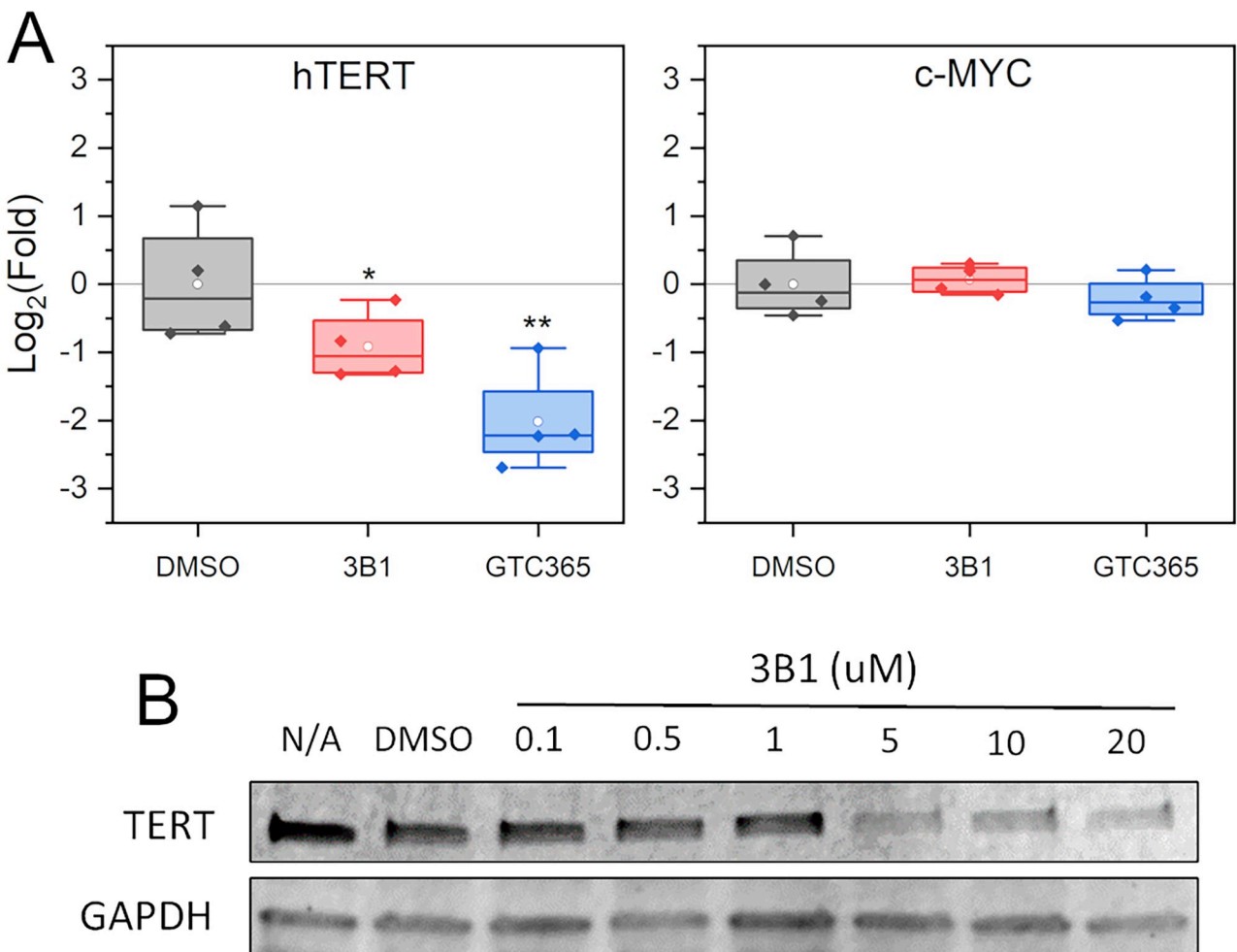

**Fig 6. Real-time quantitative PCR and immunoblotting results.** (A) qRT-PCR results of 3B1 and GTC365 (positive control) treatment of MCF7 breast cancer cells after 72hr. Media was replaced daily and contained either 10 μM compound or DMSO as control. Results show average of triplicate measurements relative to GAPDH from four independent experiments. Statistical analysis was done using a 2-Way ANOVA and Tukey post-hoc testing (alpha level of 0.05, n = 4, * = P<0.01, ** P<0.001). (B) Immunoblot analysis and densitometry of hTERT protein after 72 hr. treatment with DMSO or compound 3B1 at the given concentrations.

### 3B1 and GTC365 binding selectivity over duplex DNA

Duplex DNA is the major off-target for G-quadruplex ligands in cells, yet it is rare that ligand selectivity is rigorously assessed against alternative DNA topologies. As we've shown above, incorporating non-G4 DNA topologies early in the drug discovery pipeline can aid in filtering out non-specific scaffolds. To show that 3B1 is highly selective for hTERT G4, we measured its binding affinity by MST in the absence or presence of excess (500x) calf thymus DNA (Fig 7). 3B1's binding affinity was only marginally affected (apparent $K_d$ increase from 3 to 5 μM). However, testing the acridine-based GTC365 under identical conditions revealed a 22x reduction in apparent $K_d$ (2 to 44 μM), indicating that it has poor selectivity over duplex DNA. These results emphasize the strength of our drug discovery funnel approach in enriching for selective G4 binders.

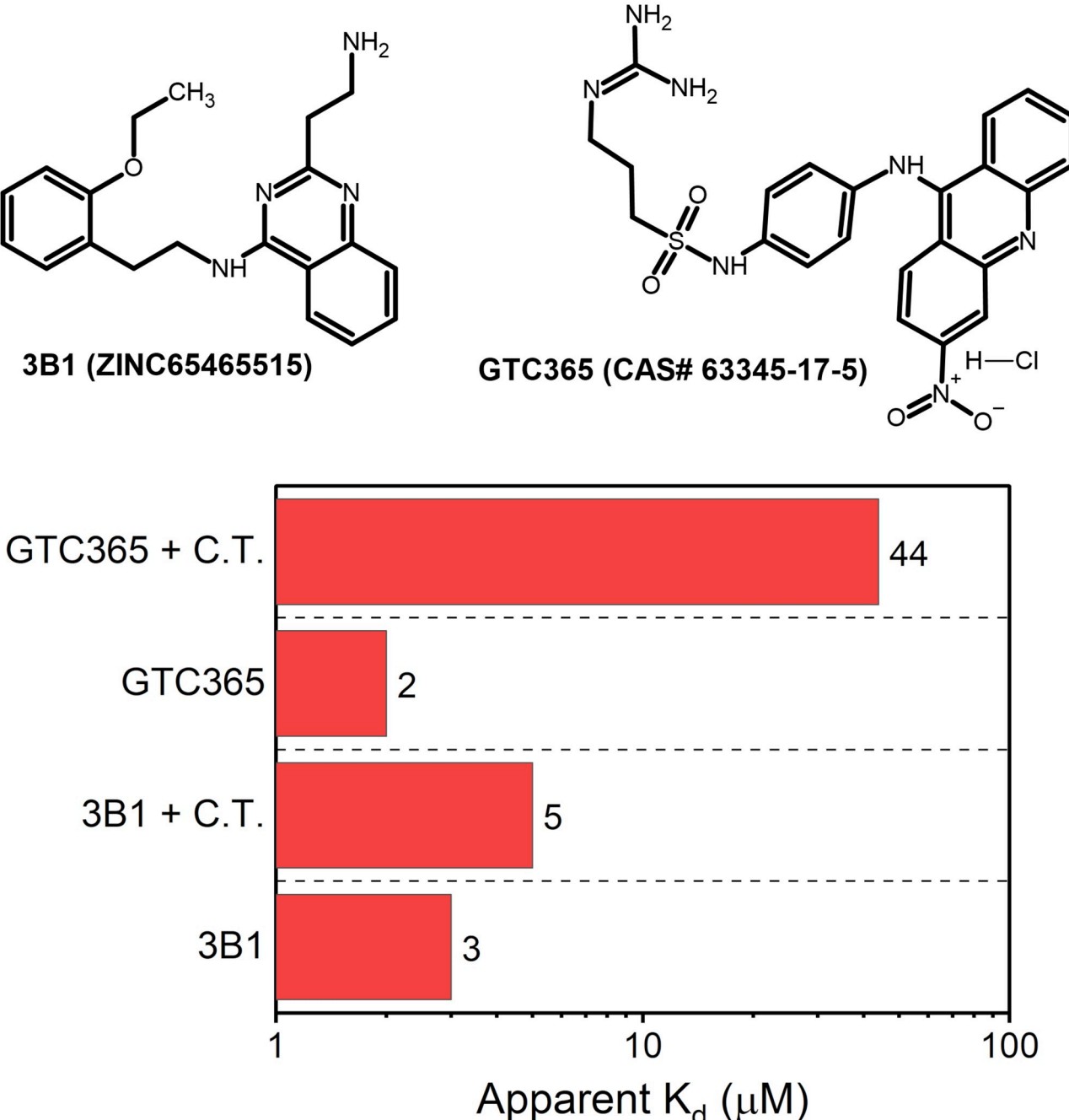

**Fig 7. Comparison of 3B1 and GTC365.** (Top) Molecular structures of compounds 3B1 and GTC365. (Bottom) Apparent dissociation constants ($K_d$) measured for 3B1 and GTC365 binding to Cy5-hTERT-FL in the presence or absence of 250 µg/mL calf thymus (C.T.) DNA (approximately 500:1 [C.T.]:[Cy5-hTERT-FL]). $K_d$ values are averages of duplicate measurements (see S8 Fig for MST traces).

## Discussion

G-quadruplexes are now widely accepted as important targets for anticancer therapeutics. To date, most attempts at targeting promoter G4s with small molecules have been limited to the small, "idealized" forms (e.g. small, often mutated, intramolecular 3-tetrad G4s). In some

cases, this has proven successful [15, 18]. However, this approach is severely limited in the following ways: (1) traditional small intramolecular G-quadruplexes often have shared characteristics such as exposed G-tetrad faces which enrich for polycyclic end-pasting molecules [76]; (2) adding to, or expanding upon core scaffolds to increase selectivity for monomeric G4s leads to a decrease in drug-likeness [25, 76], and potentially bioavailability; (3) targeting small, isolated G4s fails to take into consideration its physiological environment, such as its position within a higher-order assembly [29, 30, 36, 77]. As we have demonstrated here, these limitations can be overcome by targeting higher-order G4 systems in conjunction with the use of an expanded chemical search space and high-throughput drug discovery methods.

Enriching for drug-like small molecules requires an adequate search of chemical space [14, 78]. The only methods that can begin to approach the $\sim 10^{33}$ possible drug-like small molecules [79] are cheminformatic by nature. Here we have used an extensive multiple binding site receptor-based G4 virtual screening approach [14] targeting the unique inter-G4 junctions, loops, and grooves of the all-parallel stacked hTERT core promoter G4. Although our rate of return of compounds with acceptable affinity for hTERT is low, we believe there are two main reasons why this may be. The first is that most virtual screening software scoring functions are parameterized for proteins and, second, even if parameterized for DNA the higher-order G4 binding pockets are not canonical DNA. This opens up an opportunity for improving scoring functions with more data from higher-order quadruplex:ligand complex structures.

A major outcome of our screen is the identification of non-standard G4-interacting molecular scaffolds (Fig 1 and S2 Fig). Here "non-standard" means that very few of our initial hits share features with the common scaffolds derived from G4 virtual and actual screens or by design [14, 80, 81]. Common G4 scaffolds are flat, polycyclic, and highly conjugated molecules that often prefer interaction by stacking or end-pasting onto the terminal G-tetrads [80]. Our robust drug discovery funnel [50] approach alleviates this problem by targeting new unique features and thereby increase the likelihood of identifying useful new molecular scaffolds that aren't prone to end-pasting. An integral step in this process is our selectivity testing by competition dialysis (Fig 3). We show that compound 3B selects for hTERT over other tested DNA conformations in terms of their *monomeric unit* [73] (base pairs, triplets, or tetrads). However, G-quadruplex DNA can also be expressed in terms of the number of exposed 5' and 3' tetrad faces, as these are the sites where end-pasting occurs. hTERT has 9 contiguous G-tetrads, and hence 9 monomeric units with 2 exposed faces, whereas c-Myc, and H-Tel have 3 monomeric units (tetrads) with 2 exposed faces. Therefore, since all receptors were at the same monomeric unit concentration (75 μM), the latter sequences have 3x the concentration of exposed faces available for end-pasting, in support that 3B and its derivatives do not bind via end-pasting.

Although 3B1 binds with moderate affinity and selectivity to the higher-order hTERT G4, its effect on hTERT production in cells was relatively mild (Fig 6). The first and most obvious reason is that 3B1, which has a privileged quinazoline moiety [82], has been sequestered by off-target proteins, either in the media, cell milieu, or both. Selectivity over proteins will be addressed in subsequent work aiming to improve on 3B1 as a lead molecule. Another reason, that is less well defined and incomparable to traditional protein or duplex targeting, is the temporal dynamics of promoter G4 structures [5]. It is unclear how persistent the hTERT core promoter G4 is at any given time. However, both hTERT promoter activity and G4-formation are highest during S-phase [5, 83], which helps in explaining why 3B1 has only a modest effect after 72 hours.

The primary goal of this study was to identify small molecules with selectivity for the hTERT G4. In prior work, Kang *et al.* [38] used *in vitro* methods to screen the NCI Diversity set against the PQS23 region of the hTERT higher-order G4. In their study, the authors identified compound "GTC365", an acridine derivative with similar core scaffold as the well-known

end-pasting BRACO-19 [84]. Although significant work was done to show that GTC365 binds to the hTERT PQS23 fragment *in vitro* and *in vivo* (cells), and could also repress telomerase in breast cancer cells, the authors gave no evidence for selectivity of GTC365 over duplex DNA. BRACO-19 and other acridine scaffolds are non-selective G4 binders that also bind duplex DNA [25]. Developing selectivity for G4s is a critical first step towards quadruplex ligands as therapeutics. There are ~$5x10^5$ predicted G4 motifs in the genome [2, 85] and comparing this with the estimated number of base pairs (~$3x10^9$) indicates that there is approximately 6,000x the amount of duplex DNA to G4 DNA in a nucleus. In the presence of ~500x calf thymus duplex DNA, we showed that the previously reported GTC365 has more than a 10-fold reduction in its affinity to hTERT-FL by MST (Fig 7). Conversely, 3B1 is virtually unaffected by the presence of duplex DNA, with only a modest reduction in $K_d$ from 3 μM to 5 μM under identical conditions. Thus, even though both ligands have nearly identical binding affinities and similar intrinsic properties, 3B1 is superior in its ability to discriminate G4 over duplex DNA.

To our knowledge, this is the first report of a virtual screen against a higher-order promoter G4 system and, significantly, over 480 million compounds were docked to the 12 docking sites. Extended intramolecular G-quadruplexes are often recalcitrant to traditional structural study by NMR and X-ray diffraction. Recently, we have developed an integrative structural biology platform for characterizing higher-order G4 and G4-duplex systems at "medium" resolution [30–32]. Our approach utilizes atomistic modeling coupled with hydrodynamic, scattering, spectroscopic, and biochemical methods to construct and refine self-consistent models with uniquely targetable sites for our drug discovery funnel [50]. The hTERT core promoter (Fig 1) is composed of three parallel G4 domains, designated PQS-1, -2, and -3, arranged tail-to-head (3' on 5') to form a contiguous run of nine stacked G-tetrads. In total there are five 2-nt loops and one 4-nt loop within PQS1 and PQS2 and a 6-nt loop that bridges PQS1 and PQS2. Although multiple sites among the G4 junctional regions were targeted, site DG22, which sits within the 6-nt loop, was the top scoring site useful in rationalizing 3B1 binding (Fig 5). For reference, the SiteMap Dscore for site DG22 was 0.93, which is high considering that small monomeric G-quadruplexes (such as a parallel c-Myc G4) have scores on the order of 0.55–0.69 [30], which is considered undruggable. The stabilizing effect of 3B, 3B1, and 3B2 on PQS12 can be ascribed to the 2-aminoethyl quinazoline moiety that is predicted to bind across the phosphate backbone with the more hydrophobic end projecting into the non-polar PQS12 pocket (i.e., towards the G-tetrad column, Fig 5F and 5G). In this arrangement, multiple bridging waters facilitate the polar secondary amine (at the 4-position of the quinazoline) and ethoxy group that are projecting away from the mostly non-polar cavity (Figs 4 and 5, & S6 Fig). Although it might seem counterintuitive that so much of the binding interaction is facilitated by water bridges, a recent assessment of high resolution DNA G4 crystal structure complexes showed that the 1st and 2nd shell waters play essential roles in stabilizing ligand-G4 interactions [86]. Thus, the structural insights gained from explicit solvent MD of ligand-G4 interactions and SiteMap [74, 75] analyses are highly beneficial for future medicinal chemistry efforts. Collectively, this highlights the utility of integrative biology-derived models as receptors in virtual drug discovery approaches.

## Supporting information

**S1 Fig. CD spectra of PQS1, PQS12, PQS23, and hTERT-FL in BPEK buffer with 100 mM KCl.**
(PDF)

**S2 Fig. Structural variations of first-generation compounds (1–3) with their "SAR by catalogue" derivatives and ZINC identifiers.** Red is used to help visualize structural differences

between first-generation molecules and derivatives.
(PDF)

**S3 Fig. CD melting analysis of hTERT-FL (20 mM KCl) in the presence of BRACO-19 (2.5 μM, positive control), compounds 1N, 2R, 2S, 3A, and 3B (all at 25 μM), or an equivalent volume of DMSO (negative control).** Data are plotted as normalized molar ellipticity at 260 nm versus temperature and fit using a sigmoidal Boltzmann function to estimate the $T_m$ (plotted in the lower panel with whiskers representing estimated standard error).
(PDF)

**S4 Fig. Representative MST titrations for compound 3B and its derivatives with the Cy5-hTERT-FL G-quadruplex sequence.** In each figure, the top panel shows the raw MST traces, and the lower panels show the fit and residuals. The figures show titrations with compounds (A) 3B, (B) 3B1 as an average of quadruplicate measurements, (C) 3B2, (D) 3B3, (E) 3B4 (no reliable fit was obtained for the data), and (F) 3B5. Analysis was done in the "cold fluorescence" mode of the program PALMIST v1.5.8 [60] and fit with a 1-site binding model. Inset in each figure are the $K_d$ values, 68.3% confidence intervals in brackets, and fits (rmsd) in blue. Figures were generated in PALMIST v1.5.8.
(PDF)

**S5 Fig. Regression analysis of dissociation constant ($K_d$) with respect to melt temperature shifts ($\Delta T_m$) for (A) PQS12 and (B) PQS23. Dissociation ($K_d$) values were derived from the MST data using the hTERT-FL sequence.**
(PDF)

**S6 Fig. Interaction network and space-fill representation of 3B1 after 5ns of explicit solvent MD simulation.** (A) Hydrogen bonding network showing multiple water interactions bridging the 3B1 molecule and residues surrounding the pocket. Hydrophobic ring systems are facing inward toward the G-tetrad column. (B) Space-fill representation of hTERT-FL with 3B1 (green) shown as spherical representation to highlight the size and depth of the binding pocket.
(PDF)

**S7 Fig. Interaction network and space-fill representation of 3B1 after 5ns of explicit solvent MD simulation at site DG38.** (A) Hydrogen bonding network showing multiple water interactions, although none are coordinated with residues on the quadruplex. (B) Space-fill representation of hTERT-FL with 3B1 (green) shown as spherical representation. The two orientations emphasize how far the 3B1 ethoxy group (oxygen in red) extends away from the quadruplex.
(PDF)

**S8 Fig. MST titrations of Cy5-hTERT-FL with 3B1 and GTC365 in the presence or absence of calf thymus (C.T.) DNA.** A and C show averaged titrations of hTERT and 3B1 in the absence and presence of 250 μg/mL C.T., respectively. B and D show averaged titrations of hTERT and GTC365 in the absence and presence of 250 μg/mL C.T., respectively. Each curve was fit with a 1-site binding model in PALMIST v1.5.8 [60] using the cold fluorescence mode and the $K_d$ values, 68.3% confidence intervals, and fits (rmsd) are given in the inset in blue.
(PDF)

**S1 Raw images.**
(PDF)

## Acknowledgments

The content is solely the responsibility of the authors and does not necessarily reflect the official views of the National Institute of general Medical Sciences or the National Institutes of Health.

## Author Contributions

**Conceptualization:** Jonathan B. Chaires, John O. Trent.

**Data curation:** Robert C. Monsen, Lynn W. DeLeeuw.

**Funding acquisition:** Jonathan B. Chaires, John O. Trent.

**Investigation:** Robert C. Monsen, Lynn W. DeLeeuw, Jonathan B. Chaires, John O. Trent.

**Methodology:** Robert C. Monsen, Jon M. Maguire, Jonathan B. Chaires, John O. Trent.

**Resources:** Jon M. Maguire, Jonathan B. Chaires, John O. Trent.

**Software:** Jon M. Maguire.

**Supervision:** Jonathan B. Chaires, John O. Trent.

**Visualization:** Robert C. Monsen.

**Writing – original draft:** Robert C. Monsen.

**Writing – review & editing:** Lynn W. DeLeeuw, Jonathan B. Chaires, John O. Trent.

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
