## [Decision Letter · Decision Letter 0]

16 May 2022

PONE-D-22-09531Drug discovery of small molecules targeting the higher-order hTERT promoter G-quadruplexPLOS ONE

Dear Dr. Trent,

Thank you for submitting your manuscript to PLOS ONE. After careful consideration, we feel that it has merit but does not fully meet PLOS ONE’s publication criteria as it currently stands. Therefore, we invite you to submit a revised version of the manuscript that addresses the points raised during the review process.

Your article has been reviewed by three experts in the field. Please revise it according to comments by reviewer #2 and submit a revised version of the manuscript.

We look forward to receiving your revised manuscript.

Kind regards,

Fenfei Leng, Ph. D.

Academic Editor

PLOS ONE

Journal Requirements:

Additional Editor Comments:

Your article has been reviewed by three experts in the field. Please revise it according to comments by reviewer #2 and submit a revised version of the manuscript.

Reviewers' comments:

Reviewer's Responses to Questions

**Comments to the Author**

1. Is the manuscript technically sound, and do the data support the conclusions?

Reviewer #1: Yes

Reviewer #2: Yes

Reviewer #3: Yes

2. Has the statistical analysis been performed appropriately and rigorously? 

Reviewer #1: N/A

Reviewer #2: Yes

Reviewer #3: Yes

3. Have the authors made all data underlying the findings in their manuscript fully available?

Reviewer #1: Yes

Reviewer #2: Yes

Reviewer #3: Yes

4. Is the manuscript presented in an intelligible fashion and written in standard English?

Reviewer #1: Yes

Reviewer #2: Yes

Reviewer #3: Yes

5. Review Comments to the Author

Reviewer #1: The manuscript by Monsen et al. deals with an investigation aimed at the finding of new molecular entities that exhibit high selectivity towards G-quadruplex DNA structures over other forms of DNA. The authors have successfully combined computational and experimental techniques. In particular, they have performed an extensive virtual screen against the higher-order hTERT core promoter G-quadruplex targeting 12 of its unique loop and groove pockets using libraries containing 40 million drug-like compounds for each screen Then, using a very well consolidated discovery funnel approach, they have used high-throughput fluorescence thermal shift assay (FTSA) screens, microscale thermophoresis (MST), and two rounds of catalogue-based SAR to find a disubstituted 2-aminoethyl-quinazoline that stabilizes the higher-order hTERT G-quadruplex by binding across its G4 junctional sites, exerting a very interesting biological activity. The manuscript is well written and scientifically very convincing. For these reasons the manuscript deserves to be published as it is.

Reviewer #2: What are the main claims of the paper and how significant are they for the discipline?

The authors define a problem with most previous G4 drug design efforts as the use of inappropriate, small model systems without appropriate druggable sites. They propose a new model system and approach that can change the way drugs are discovered for G4 systems (see comments to authors - below)

Are the claims properly placed in the context of the previous literature? Have the authors treated the literature fairly?

Yes to both. See the Introduction to the paper.

Do the data and analyses fully support the claims? If not, what other evidence is required?

The data, results, and analyses are clearly and thoroughly presented and fully support the conclusion in the paper. Appropriate error analysis is reported for each section.

PLOS ONE encourages authors to publish detailed protocols and algorithms as supporting information online. Do any particular methods used in the manuscript warrant such treatment? If a protocol is already provided, for example for a randomized controlled trial, are there any important deviations from it? If so, have the authors explained adequately why the deviations occurred?

Methods, protocols, and other details are either described in the paper or referenced.

If the paper is considered unsuitable for publication in its present form, does the study itself show sufficient potential that the authors should be encouraged to resubmit a revised version?

The paper is essentially publishable as is – see minor comments in the comments to authors

Are original data deposited in appropriate repositories and accession/version numbers provided for genes, proteins, mutants, diseases, etc.?

Yes

Does the study conform to any relevant guidelines such as CONSORT, MIAME, QUORUM, STROBE, and the Fort Lauderdale agreement?

NA

Are details of the methodology sufficient to allow the experiments to be reproduced?

Yes

Is any software created by the authors freely available?

NA

Is the manuscript well organized and written clearly enough to be accessible to non-specialists?

Yes – highly readable

Is it your opinion that this manuscript contains an NIH-defined experiment of Dual Use concern?

No

Review Comments:

In the Introduction, the authors clearly define the significance and importance of DNA G-quadruplexes (G4) for cell physiology and, in this work, as potential drug targets. They note the lack of success in the search for successful compounds for clinical trials with G4s as a target. They logically suggest that the primary reason for this lack of success is due to the G4 drug design models in most current research. These models generally use monomeric G4 structures that are relatively small and do not have appropriate druggable sites. Most compounds selected to bind G4s to date interact by an end-stacking mode that can provide strong binding to the G4 model but, unfortunately, limited selectivity in binding. In a very innovative approach, they propose to correct this problem with the use of higher-order, larger models to provide more drug-like targeting sites on G4 structures. This approach should allow strong binding with much-improved selectivity, a feature generally lacking in previous G4 compounds. They have previously developed a multi-G4 model with the hTERT core promoter G4 and it is used as the model system in this report. They apply an appropriate virtual screening approach with over 40 million small molecules to search for successful drug candidates with the hTERT model system. This approach is particularly significant considering the current modest success with smaller models and has the potential to establish an entirely new method for the discovery of G4 targeting drugs. The hTERT G4 model used is shown in Figure 1. The new approach used by the authors targets 12 potentially druggable docking sites chosen by targeting loops, grooves, and G4 junctions in the model. These sites, also displayed in Figure 1, are quite different than those used in previous research. From the screening, 100 optimum scaffolds were defined and 69 of these were commercially available.

Tm methods were used to test the 69 compounds with hTERT models and two compounds were found to effectively stabilize at least one of the hTERT Tm models at appropriate buffer/salt conditions. Another compound was found to give lower but acceptable stabilization. It is reassuring that the three compounds did not significantly stabilize duplex DNA. The three compounds (1-3) have not been previously reported as G4 stabilizers. To me, they look quite different than any other reported G4 binding agents. The low rate of return of compounds with acceptable affinity for hTERT, however, raises a concern with the screening set. A better G4/nucleic acid set of molecules appears to be an important future area for development for G4 drug discovery. The authors next searched a commercial catalog for compounds similar to 1-3 and obtained 25 additional test compounds. Of these, six compounds gave significant stabilization of an hTERT model and only one gave appreciable DNA duplex binding. These five are an encouraging breakthrough in the hTERT and extended G4 drug design area. They provided important new ideas for the design of compounds to selectively target G4s.

The authors next investigated the selectivity of the compounds chosen based on 1-3 by competition dialysis. This method was largely developed by Chaires and uses direct comparisons of selectivity to multiple different DNA structures. The results show that compounds 3B and 2S (Figure S2) have a strong preference for G-quadruplex DNA

over duplex and triplex conformations and have a strong preference for the hTERT model over other G4 structures. The studies are rigorously done with appropriate error analysis. Based on the successful results with compound 3, additional structures were selected from available agents and compounds 3B1 and 3B2 had improved thermal stabilizing results with good selectivity. More direct binding studies indicated the strongest binding to the hTERT model by 3B1.

While the experimental studies provide convincing evidence of the strong binding of Compound 3 and analogs to the hTERT models, they do not provide evidence of how and where the compounds bind to the G4 structure. To answer this question molecular modeling studies were carried out using their hTERT structure. Fortunately, this is an area of expertise by the Trent group. The results shown in Figure 5 provide strong evidence for strong binding of 3B,3B1, and 3B2 to two specific sites on the hTERT model. Analysis of the bound systems provides strong evidence for possible interactions in the hTERT complex. Appropriate controls, statistical tests, and error analysis were done in this and other studies reported in this paper.

Two additional very important studies that are rarely done set this paper well ahead of most others in this field: (i) In Figure 6 the authors show that compound 3B1 targets hTERT in cancer cells but also that it does not target another well-characterized G4, c-MYC; (ii) in Figure 7 the authors look at binding of 3B1 and an acridine compound found to target hTERT – then they look at binding of the compounds when they are challenged with a large excess of duplex DNA – binding of 3B1 is only slightly decreased, indicating excellent selectivity while the activity of the acridine is decreased over 20 times. The selectivity test is essential for the evaluation of new compounds and should be a requirement for publication of proposed new G4 agents.

Minor Point: The activity for 3B1 in cancer cells (Figure 6) seems relatively low, given its binding affinity and selectivity for hTERT. It would be useful for the authors to briefly address this point.

Reviewer #3: This is a well written solid manuscript discussing drug discovery for small molecules to target the hTERT sequence. A variety of experimental and computational studies were carried out. The results were presented in a logical systematic fashion and the conclusions drawn reasonable. I see no need for any revision.

6. PLOS authors have the option to publish the peer review history of their article (what does this mean?). If published, this will include your full peer review and any attached files.

Reviewer #1: No

Reviewer #2: **Yes: **David Wilson

Reviewer #3: No

---

## [Author Response · Author response to Decision Letter 0]

31 May 2022

Dear Dr. Fenfei Leng,

Thank you for your letter informing us of the PLOS ONE decision. We have uploaded the minor revision requested in the format that was requested. We have considered and addressed all comments by the reviewers and the editor:

Reviewer 1 Comment:

… The manuscript is well written and scientifically very convincing. For these reasons the manuscript deserves to be published as it is.

Response: No revisions were requested. We thank the reviewer for their enthusiastic support.

Reviewer 2 provided highly complimentary and positive comments. Only two comments were made that required action and revision. The two comments are addressed below.

Reviewer 2 Comment:

The low rate of return of compounds with acceptable affinity for hTERT, however, raises a concern with the screening set. A better G4/nucleic acid set of molecules appears to be an important future area for development for G4 drug discovery. 

Response: Additional text added

“Although our rate of return of compounds with acceptable affinity for hTERT is low, we believe there are two main reasons why this may be. The first is that most virtual screening software scoring functions are parameterized for proteins and, second, even if parameterized for DNA the higher-order G4 binding pockets are not canonical DNA. This opens up an opportunity for improving scoring functions with more data from higher-order quadruplex:ligand complex structures.” 

This was an excellent point to make and it really justifies more investigations such as ours as we need more validated higher-order DNA:ligand complexes to be able to parameterize such screening efforts to be more efficient. This is the first report of higher-order DNA:ligand screening effort, so therefore we could not parameterize the scoring functions as no structures exist. 

Reviewer 2 Comment:

Minor Point: The activity for 3B1 in cancer cells (Figure 6) seems relatively low, given its binding affinity and selectivity for hTERT. It would be useful for the authors to briefly address this point.

Response: Additional text added

“Although 3B1 binds with moderate affinity and selectivity to the higher-order hTERT G4, its effect on hTERT production in cells was relatively mild (Figure 6). The first and most obvious reason is that 3B1, which has a privileged quinazoline moiety(83), has been sequestered by off-target proteins, either in the media, cell milieu, or both. Selectivity over proteins will be addressed in subsequent work aiming to improve on 3B1 as a lead molecule. Another reason, that is less well defined and incomparable to traditional protein or duplex targeting, is the temporal dynamics of promoter G4 structures(5). It is unclear how persistent the hTERT core promoter G4 is at any given time. However, both hTERT promoter activity and G4-formation are highest during S-phase(5, 84), which helps in explaining why 3B1 has only a modest effect after 72 hours.”

Reviewer 3 Comment:

This is a well written solid manuscript discussing drug discovery for small molecules to target the hTERT sequence. A variety of experimental and computational studies were carried out. The results were presented in a logical systematic fashion and the conclusions drawn reasonable. I see no need for any revision.

Response: No revisions were requested. We thank the reviewer for their enthusiastic support.

Editor’s Comment:

Your article has been reviewed by three experts in the field. Please revise it according to comments by reviewer #2 and submit a revised version of the manuscript.

Response: We have done so. Reviewer 2 had only two comments that required action as addressed above.

We have fully addressed all of Reviewer 2 comments.

---

## [Editor Report · Decision Letter 1]

6 Jun 2022

Drug discovery of small molecules targeting the higher-order hTERT promoter G-quadruplex

PONE-D-22-09531R1

Dear Dr. Trent,

We’re pleased to inform you that your manuscript has been judged scientifically suitable for publication and will be formally accepted for publication once it meets all outstanding technical requirements.

Kind regards,

Fenfei Leng, Ph. D.

Academic Editor

PLOS ONE
---

## [Editor Report · Acceptance letter]

8 Jun 2022

PONE-D-22-09531R1 

Drug discovery of small molecules targeting the higher-order hTERT promoter G-quadruplex 

Dear Dr. Trent:

I'm pleased to inform you that your manuscript has been deemed suitable for publication in PLOS ONE. Congratulations! Your manuscript is now with our production department. 

Kind regards, 

on behalf of

Dr. Fenfei Leng 

Academic Editor

PLOS ONE